# Waste from *Persea schiedeana* Fruits as Potential Alternative for Biodiesel Production

**DOI:** 10.3390/plants11030252

**Published:** 2022-01-19

**Authors:** Anallely López-Yerena, Diana Guerra-Ramírez, Benito Reyes-Trejo, Irma Salgado-Escobar, Juan Guillermo Cruz-Castillo

**Affiliations:** 1Department of Nutrition, Food Science and Gastronomy, XIA, Faculty of Pharmacy and Food Sciences, Institute of Nutrition and Food Safety (INSA-UB), University of Barcelona, 08028 Barcelona, Spain; naye.yerena@gmail.com; 2Laboratorio de Productos Naturales, Área de Química, Departamento de Preparatoria Agrícola, Universidad Autónoma Chapingo, Km 38.5 Carretera México-Texcoco, Chapingo 56230, Mexico; dguerrar@chapingo.mx (D.G.-R.); breyest@chapingo.mx (B.R.-T.); 3Escuela de Ingeniería y Ciencias, Departamento de Ciencias, Campus Ciudad de México, Tecnológico de Monterrey, Calle del Puente 222, Ejidos de Huipulco, Tlalpan, Mexico City 14380, Mexico; isalgado@tec.mx; 4Centro Regional Universitario Oriente (CRUO), Universidad Autónoma Chapingo, Km. 6, Carretera Huatusco—Jalapa, en Huatusco, Veracruz 94100, Mexico

**Keywords:** green chemistry, feedstock, underutilized fruit trees, added value, waste fruits, *Heilipus lauri*

## Abstract

Biodiesel is a mixture of monoalkyl esters of fatty acids derived from vegetable oils or animal fats. Agricultural residues are a potential source of raw materials for generating biofuels. The aim of this research was to determine the usefulness of *Persea schiedeana* Nees fruit as a potential source to be transformed into biodiesel by alkaline transesterification. In this sense, biodiesel was obtained using oil isolated from overripe fruits of *P. schiedeana*, damaged by the *Heilipus lauri* pest. The fruits were collected in the municipality of Huatusco, Veracruz, Mexico. The maceration of the fruits with hexane resulted in an oil with a high percentage of free fatty acids (8.36 ± 1.35%). The main components of the biodiesel were methyl oleate (53.12%) and methyl palmitate (25.74%). The dynamic viscosity of the biodiesel complies with ASTM D6751 and EN 14214 and the biodiesel blends with commercial diesel comply with ASTM D675, the calorific value showed an increase with increasing biodiesel concentration in the blends. This study demonstrates that the oil obtained from the overripe and surplus fruits of *P. schiedeana* is a viable feedstock for the production of a fuel to replace diesel.

## 1. Introduction

There is currently a growing interest in researching wild fruits to find their possible uses and health benefits and to promote their consumption and production. *Persea schiedeana* Nees is a fruit tree distributed from central and southern Mexico to Panama on a variety of soil types in forests and tropical lands from 90 to 2000 m of altitude [1,2]. *P. schiedeana* has a recognized value as a consumer fruit with the opportunity to increase its production to stimulate local economies through fresh sales as well as its agro-industrial use with the extraction of oil from the pulp [3], using technologies such as centrifugation [1,4]. However, the fruits are attacked by the borer *Heilipus lauri* from the first days after set, causing the abortion of the fruits; in addition, the plague attacks during the whole cycle of fruit development cause considerable losses [5]. Another factor that has limited its commercial development has been its short post-harvest life (8 days), being a very perishable fruit with a high respiration rate [6].

The reduction in fossil fuels and the environmental impact caused by their use are the main reasons for the search for alternative fuels from renewable resources. Biodiesel, an alternative to diesel fuel, is obtained from renewable biological sources such as vegetable oils and animal fats [7].The need to reduce greenhouse gas emissions has increased the demand for vegetable oils in the food, chemical, and biodiesel industries [8]. Interest in the use of renewable fuels began with the direct use of vegetable oils as a substitute for diesel [9]. Biodiesel, a common term for fatty acid methyl esters (FAMEs), has been hailed as a renewable, biodegradable biofuel resource that is non-toxic and holds great promise for diesel fuel replacement [10].

The most widely used process to produce biodiesel to date has been transesterification, which is a reaction between an oil and an alcohol to obtain monoalkyl esters of fatty acids (biodiesel) and glycerin [11]. Two of the most important thermophysical properties of biodiesel are viscosity and specific heat [12]. Absolute viscosity is defined as the resistance to flow and shear due to internal friction. On the other hand, kinematic viscosity is the resistance to flow and shear due to gravity [13]. Biodiesel is considered the best substitute for diesel and can be used pure or in biodiesel/diesel blends, offering environmental, economic, social, and technical advantages [14].

Due to the significant growth in the population, the increase in energy demand, and the fact that fossil fuels are limited sources of energy, to date, some authors have tried to examine the potential of food waste [15,16] such as avocado seed as an environmentally friendly alternative energy source [17,18,19]. However, the potential of *P. schiedeana* fruits for biodiesel production has not been evaluated so far despite the considerable losses of these fruits attributed to their short postharvest life and pest attacks. Thus, the aim of this research was to determine, for the first time, the usefulness of *P. schiedeana* fruits (overripe, dehydrated and with the presence of pests) as a potential source to be transformed into biodiesel by alkaline transesterification.

## 2. Results

### 2.1. Oil Extraction by Maceration

The oil yield obtained by maceration of the fruits of *P. schiedeana* was 93.23%, compared to the exhaustive extraction process by Soxhlet (24.99 ± 1.14%).

### 2.2. Chemical Properties of the Oil

The oil was characterized by a high percentage of free fatty acids (FFA, 8.36 ± 1.35%), which is higher than that specified by the standard for avocado oil (1.5%). The oxidation process of the FFA and the formation of free radicals in *P. schiedeana* oil was 3.99 ± 0.58 O_2_·kg^−1^, lower than the specification allowed for avocado oil. In addition, the oil presented a lower degree of unsaturation (75.05 ± 1.09 g I_2_ 100·g^−1^) than the range specified in the Mexican standard for avocado oil. Finally, the saponification value, used to estimate the chain length of the fatty acids, was 179.52 ± 2.85 mg KOH·g^−1^, which complies with the specification allowed for avocado oil (Table 1).

### 2.3. Biodiesel Properties

The conversion of *P. schiedeana* oil to biodiesel, performed with alkaline transesterification after pretreatment with sulfuric acid, reached 100% conversion. Figure 1 shows the hydrogen nuclear magnetic resonance (^1^H NMR) spectrum of the biodiesel obtained by the transesterification reaction of *P. schiedeana* oil.

#### 2.3.1. Fatty Acid Profile

The fatty acid profile of *P. schiedeana* oil is presented in Table 2. The highest proportion of fatty acids quantified corresponded to monounsaturated fatty acids (MUFAs, 58.74%), of which 9(Z)-octadecenoic acid was the main one (53.12%), followed by 9(Z)-Hexadecenoic (5.62%), and a low presence of polyunsaturated fatty acids (PUFAs) such as 9,12(Z,Z)-Octadecadienoic and 9,12,15(Z,Z,Z)-Octadecatrienoic (4.84 and 0.57%, respectively).

#### 2.3.2. Mechanical Properties and Calorific Value of Oil and Biodiesel

The density of the oil, diesel, biodiesel, and their respective blends (B5, B10, B20, B30 and B40) are shown in Figure 2A, where when increasing the temperature in all the treatments, the density decreased.

The kinematic viscosity in *P. schiedeana* oil obtained by maceration decreased with increasing temperature from 20–90 °C (106.19–11.08 mm^2^·s^−1^). The viscosity value of vegetable oil methyl esters decreased considerably after transesterification [9]. The viscosity of *P. schiedeana* biodiesel was 4.50 mm^2^·s^−1^ at 40 °C, indicating that the produced biodiesel had a low viscosity, which means the biodiesel could flow easily (Figure 2B). The dynamic viscosity of *P. schiedeana* oil was in the range of 88.90 to 8.14 mPas^−1^ from 20 to 90 °C. Finally, the oil, the biodiesel, and the blends prepared with commercial diesel were characterized with the calorific value (MJ·kg^−1^) as is shown in Figure 2C.

## 3. Discussion

Claims about the search for alternatives for fossil fuels have left the potential use of food waste to produce biodiesel. In this study, for the first time, we evaluated the use of under-utilized fruits (overripe, dehydrated, and with the presence of pests) of *P. schiedeana* as alternative for biodiesel production.

The yield of oil obtained by maceration (23.29%) was higher than that of the oil obtained from overripe *P. schiedeana* fruits by maceration (12.12%) with the isopropanol/hexane mixture [20]. This yield was also higher than that obtained by Joaquín-Martínez et al., [22] in fruits from the region of Los Tuxtlas, Veracruz (24.7–36%). These results suggested that the use of hexane for oil extraction could increase recovery values.

Regarding the chemical properties of the oil, to date, there are very few studies on the chemical characteristics of the *P. schiedeana* oil. In 2011, Campos-Hernandez et al. found a lower percentage of FFA (0.16%) in oil obtained from overripe fruits by maceration [20]. Differences in the results can be attributed to the fact that in the aforementioned study they added butylated hydroxyanisole (BHA) to the pulp to avoid the oxidation of the oil during extraction by maceration.

According to the oxidation process of the FFA and the formation of free radicals, our results suggest that the oil presented low oxidation due to the low proportion of linolenic acid (0.56%), being less susceptible to the appearance of undesirable odors and flavors caused by the formation of aldehydes, alcohols, and ketones caused by hydroperoxides and peroxides [23]. The susceptibility of fatty acids to oxidation is directly related to their degree of unsaturation [24,25]. *P. schiedeana* oil with greater oxidation was described by Campos-Hernández et al. (2011) in oil obtained from overripe fruits as shown in Table 1 [20]. In addition, the oil described in previous studies presented a higher degree of unsaturation than that obtained in this study as shown in Table 1. The oil analyzed by Campos-Hernández et al. [20] presented higher saponification value than that obtained in this study (197.13 mg KOH·g^−1^).

### 3.1. Biodiesel Chemical Characterization

Alcoholysis of vegetable oils depends on temperature, reaction time, alcohol:vegetable oil molar ratio, alcohol type, catalyst type and concentration, mixing intensity, FFA, and moisture content [26]. The ^1^H NMR spectrum of *P. schiedeana* biodiesel shows the characteristic signals of FAMEs [27]. At 5.35 ppm, a multiple signal of olefinic protons was observed; at δ 3.67, a single signal was observed corresponding to the protons of the methoxy group; and between 2.77 and 2.80 ppm, a triple signal of bialylic protons was observed. Methylenes adjacent to the ester group were observed as a multiple signal between 2.30 and 2.33, and multiple allylic protons were observed as a signal at 2.00 ppm; between 1.60 and 1.65 ppm, the signal of the methylene located at C-3 of the chain was observed. Signals in the range of 1.27–1.38 are due to other methylene protons of the fatty acid ester chain, and at the δ 0.88–0.91 terminal, methyl protons were observed as a triple signal.

The influence of fatty acid composition on the oxidation rate of biodiesel is greater than the influence of environmental conditions such as light, air and the presence of metals [28,29]. Methyl esters of MUFAs, such as 18:1, are considered better than polyunsaturated fatty acids (PUFAs) such as methyl linoleate (18:2) and linolenate (18:3) in terms of oxidative stability [30]. Oxidation stability is one of the main problems affecting the use of biodiesel due to its high content of polyunsaturated methyl esters [31]. Compared with other studies, the fatty acid profile was similar to that obtained by Cruz-Castillo et al. [22]. On the other hand, *P. schiedeana* oil with lower 9(Z)-Octadecenoic has also been detected [32]. Compared with our results, higher presence of 9(Z)-Octadecenoic and PUFAs (58.6 and 10.8%, respectively) has been demonstrated for avocado oil [27]. Oils that include high percentages of MUFAs methyl esters of 16 to 18 carbons are ideal for conversion to biodiesel [30]. According to our results, the *P. schiedeana* oil obtained from overripe fruits and with presence of the *Heilipus lauri* pest, presented a higher proportion of MUFAs; therefore, it is a viable source for obtaining biodiesel.

### 3.2. Biodiesel Physical Characterization

The density of the oil complies with the specification allowed by NMX-F-052-SCFI-2008 for avocado oil (0.910 to 0.920 g·cm^−3^) [21]. Likewise, at 20 °C, the oil is in the range for liquid oils (0.909–0.921 g·cm^−3^) [33]. The density of liquid oils is dependent on the composition of their fatty acids [34], but also on the time and temperature to which they are subjected; it decreases linearly with increasing temperature [34]. The biodiesel obtained from *P. schiedeana* fruits complies with the specification allowed by International Standard (EN 14214 2003, method IS0 3675, ISO 12185) [35] from 860 to 900 kg·m^−3^ at 15 °C. Similar results have been demonstrated for avocado biodiesel (0.875 g·cm^−3^) [36] and avocado seed biodiesel (0.86 g·mL^−1^) [19].

The main problem associated with the use of pure vegetable oils as diesel fuel is their high viscosity on compression. Viscosity is one of the most important parameters influencing machine performance. It represents a basic design specification for fuel injectors used in diesel engines, and if it is too high, it causes the injectors to malfunction [17]. Thus, triglycerides with low molecular weight have lower viscosity than those typically found in conventional vegetable oil [30]. The kinematic viscosity of biodiesel and the different blends (Figure 2B) complies with the Standard Specification (ASTM D 6571-02, method D445) of 1.9–6.0 mm^2^·s^−1^ at 40 °C [37]. In addition, the biodiesel complies with the specification allowed by International Standard (EN 14214 2003, method IS0 3104) of 3.5–5.0 mm^2^·s^−1^ at 40 °C [35]. Lower viscosity values have been demonstrated for avocado biodiesel (4.42 mm^2^·s^−1^ and 3.75 mm^2^·s^−1^) [27,38] and avocado seed biodiesel and 3.94 and 4.42 mm^2^·s^−1^ [19]. On the other hand, another study demonstrated a higher value of viscosity for avocado seed biodiesel (5.02 mm^2^·s^−1^) [18].

The dynamic viscosity of *P. schiedeana* oil was lower than that described by Diamante and Lan (2014) in avocado oil at 50 °C (28.7 mPas^−1^) [13]. Changes in viscosity with increasing temperature can be explained by the fact that as the biodiesel cools, the ester chains lose mobility, complicating flow and thus increasing the dynamic viscosity or fluid shear strength.

Finally, the oil showed lower heat of combustion than biodiesel and as the concentration of biodiesel in the blends increased, the heating value decreased. The heat of combustion of vegetable oils is higher when they have a higher amount of long-chain fatty acids and increases according to the length of the chain [39,40]. In addition, the presence of chemically bound oxygen accounts for the lower calorific value of biodiesel [41]. Our results are in line with those demonstrated for avocado biodiesel where higher heat of combustion was detected in biodiesel than in oil (40.106 and 39.485 kJ·kg^−1^, respectively) [36]. Similar heating value was also demonstrated for avocado biodiesel (40.365 kJ·kg^−1^) [38].

## 4. Materials and Methods

### 4.1. Chemicals and Reagents

Hexane, methanol, sulfuric acid, anhydrous sodium sulfate, potassium hydroxide, citric acid, deuterated chloroform (CDCl_3_), and tetramethylsilane (TMS) were purchased from Merck-Sigma Aldrich Company. Mexico City, Mexico.

### 4.2. Plant Material

The fruits of *P. schiedeana* were randomly collected from eight seedlings of about 20 years old in August 2015. The trees, which were 15 to 20 m tall, were in wild conditions. They were located at 1300 m altitude in Huatusco, Veracruz, Mexico (19° 9′14.8″ N, 96° 58″ W) in a lithosol soil. The annual average temperature was 18 °C and rainfall was 1700 mm. The fruits harvested were overripe, with 14.89% humidity, and infested with *Heiliipus lauri*.

### 4.3. Oil Extraction

The pulp and peel of *P. schiedeana* fruits were cut into small pieces and placed in a jar. The plant material was completely covered with hexane and allowed to stand during 48 h. At the end, the solvent was removed at a rotary evaporator (Büchi Rotovapor, Büchi Labortechnik AG, Flawil, Switzerland) applying vacuum and in a steam bath at 40 °C. To optimize oil extraction, the extraction process was repeated two more times. The oil obtained was weighed and placed in an amber flask until analysis and the percentage of oil obtained was calculated according to Equation (1).
(1)$$Oil\;content\;\left(\%\right)=\frac{\left(100-M\right)\;*\;W}{m}$$
where *M* is the moisture content (%), *W* is the weight of the oil extracted (g), and *m* is the weight of the pulp sample (g).

The percentage of recovered oil obtained by maceration was calculated according to Equation (2).
(2)$$Recovered\;oil\;\left(\%\right)=\frac{\left(100*m_{2}\right)}{m_{1}}$$
where *m*_1_ is the oil obtained by Soxhlet according the methodology proposed previously [4] and *m*_2_ the oil obtained by maceration.

### 4.4. Oil Analysis

#### 4.4.1. Chemical Properties of the Oil

The parameters, acid index, peroxides, iodine, and saponification were analyzed in triplicate, according to the methods of the International Organization for Standardization (ISO): acid number (ISO 660:2009) [42], saponification number (ISO 3657:2013) [43], peroxides number (ISO 3960:2007) [44], and iodine value (ISO 3961:2013) [45]. The determinations were performed in triplicate.

#### 4.4.2. Fatty Acid Profile

The oil obtained by maceration presented a high percentage of free fatty acids (8.36 ± 1.35%); therefore, it was subjected to an acid esterification reaction following the methodology of Zhang and Jiang [46] with modifications. The oil (100 g) was mixed with 100 mL of methanol-sulfuric acid (96:4 *v*/*v*). The mixture was placed under reflux and constant stirring during 2 h at 60 °C. After the reaction time had elapsed, the mixture was placed in a separatory funnel and the upper part of the methanol-sulfuric acid was removed. Finally, the oil was dried with anhydrous sodium sulfate.

After pretreatment, the oil was subjected to an alkaline transesterification reaction according to Marroquín-Andrade et al. [47]. The reaction mixture 0.25:1 methanol/oil (*w*/*w*) and 0.7% of potassium hydroxide by weight of oil was placed under reflux and constant stirring for 2 h. After the time elapsed, the reaction mixture was transferred to a separatory funnel and left for 24 h until phase separation was observed, the upper phase consisting of the FAMEs and the lower phase of glycerol.

After discarding the glycerol phase, the FAMEs were subsequently treated with 0.1% citric acid and hot water. Finally, the FAMEs were dried with anhydrous sodium sulfate. The NMR spectra of the FAMEs were recorded on an Agilent spectrophotometer operated at 400 MHz (^1^H NMR). The sample concentration used was 5–10% in deuterated chloroform (CDCl_3_) containing tetramethylsilane (TMS) as internal reference.

The FAMEs were analyzed using an Agilent 6890 gas chromatograph with a flame ionization detector (FID) using an ATSilar column (30 m × 0.25 mm i.d. ×0.25 μm film thickness). The initial oven temperature was 170 °C with 10 °C ramps set per minute to a final temperature of 240 °C. Both the injector and detector temperatures were 260 °C. Hydrogen was used as carrier gas at a flow rate of 1.8 mL min^−1^. Standard FAMEs were used to quantify the FAMEs in the sample.

#### 4.4.3. Physical Properties of Oil and Biodiesel

The biodiesel obtained from the *P. schiedeana* fruits was blended at different concentrations with commercial diesel (0, 5, 10, 20, 30, 40, and 100% biodiesel). Dynamic and kinematic viscosity as well as density were determined for oil, biodiesel, and diesel/biodiesel blends in different proportions in a viscometer (Anton Paar model SVM 3000, Stabinger type, dual viscometer and density meter, according to ASTM D7042). The heat of combustion of the oil, biodiesel, and diesel/biodiesel blends were determined using an isoperybolic calorimeter according to ASTM D240 method. The values were expressed in MJ·kg^−1^.

## 5. Conclusions

The oil obtained by maceration of overripe *P. schiedeana* fruits (pulp and peel) infested with *Heilipus lauri* allowed the recovery of 93.23% of the total available oils, which presented formation of free fatty acids (8.36 ± 1.35%). Biodiesel was obtained by alkaline transesterification after acid pretreatment.

The composition of FAMEs showed a higher proportion of MUFAs; therefore, it is a viable source for obtaining biodiesel. The density of the oil and biodiesel meet the specification allowed by the Mexican Standard for avocado oil and the International Standard for biodiesel. The kinematic viscosity of the oil decreased considerably after transesterification and the viscosity of the biodiesel met the specification allowed by ASTM D 6571-02 for pure biodiesel. Oil showed lower heat of combustion than biodiesel, and as the concentration of biodiesel in the blends increased, the heating value decreased.

This is the first study on physicochemical characterization of *P. schiedana* biodiesel, and it provides the baseline of future work on potential use of waste fruits for biodiesel production. More studies are needed to optimize the extraction conditions.

## Figures and Tables

**Figure 1 plants-11-00252-f001:**
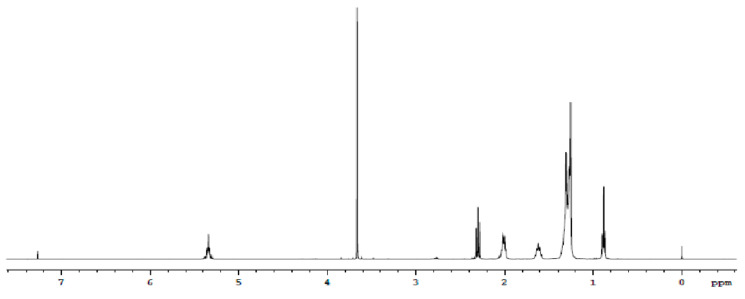
^1^H NMR spectrum of *P. schiedeana* oil methyl esters.

**Figure 2 plants-11-00252-f002:**
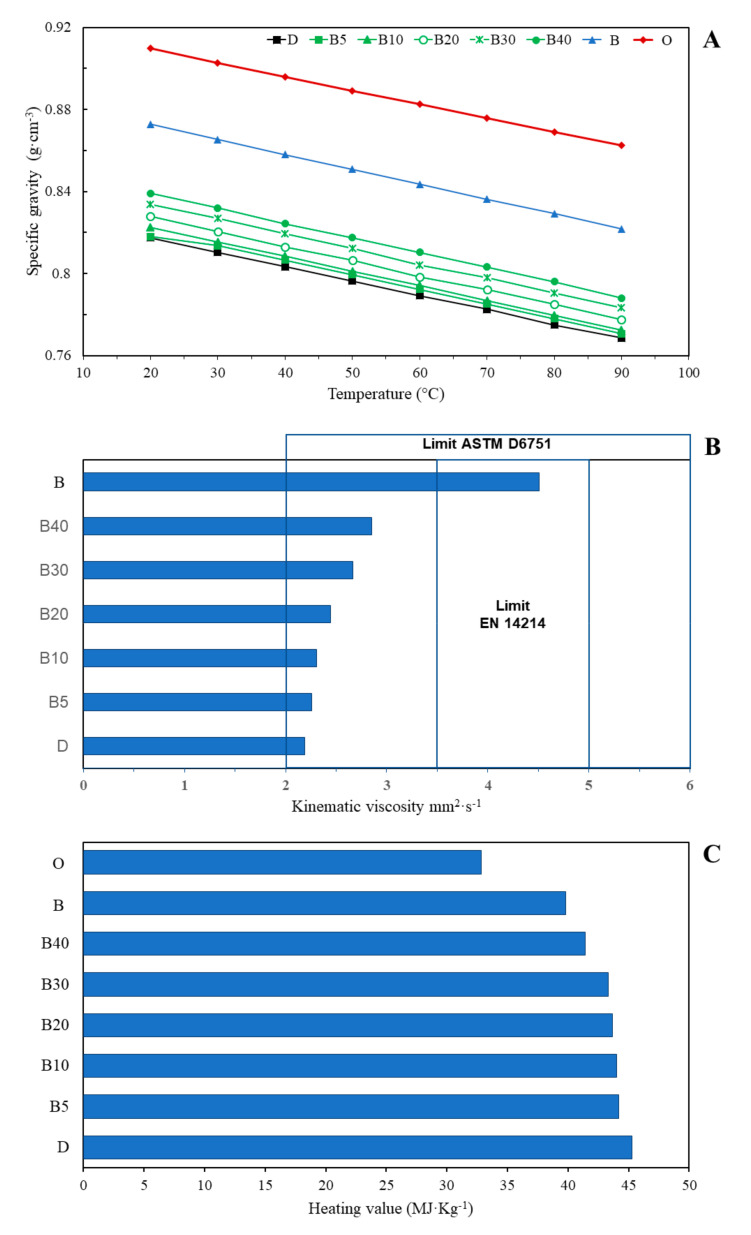
Specific gravity (**A**), kinematic viscosity (**B**), and heating value of *P. schiedeana* Nees, biodiesel, diesel, and its blends (**C**).

**Table 1 plants-11-00252-t001:** Chemical properties of the oil obtained by maceration of *P. schiedeana* fruits. Reported data are also included.

FFA (%)	Iodine (g I_2_ 100·g^−1^)	Peroxides (meq O_2_·kg^−1^)	Saponification (mg KOH·g^−1^)	Study
8.36 ± 1.35	75.05 ± 1.09	3.99 ± 0.58	179.52 ± 2.85	Current study
0.16	84.13	9.67	197.13	[20]
1.5	85–90	10	177-198	[21]

FFA: Free fatty acids. Results are expressed as the mean ± standard deviation.

**Table 2 plants-11-00252-t002:** Fatty acid profile of *P. schiedeana* oil.

Fatty Acid	%
Hexadecanoic (palmitic; 16:0)	25.75
9(Z)-Hexadecenoic (palmitoleic; 16:1)	5.62
Octadecanoic (stearic; 18:0)	3.31
9(Z)-Octadecenoic (oleic; 18:1)	53.12
9,12(Z,Z)-Octadecadienoic (linoleic; 18:2)	4.84
9,12,15(Z,Z,Z)-Octadecatrienoic (linolenic; 18:3)	0.57
Docosanoic acid (behenic; 22:0)	2.24
Others	4.57

## Data Availability

The data presented in this study are available in article.

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
