# Peer review of "Waste from *Persea schiedeana* Fruits as Potential Alternative for Biodiesel Production"

_plants, 2022, doi:10.3390/plants11030252_

Round 1
Reviewer 1 Report
Idea of the research of the manuscript is interesting and relevant, however the manuscript is weak. The authors' contribution is not significant, as they did not optimize the biodiesel production process, but applied the optimal conditions chosen by other authors. However, it cannot be said that these conditions are the optimum because other authors used different raw materials (Zanthoxylum bungeanum and ilama seeds oils). If others authors’ obtained optimum conditions are used, esters must be tested for compliance with standards EN14214 and / or ASTM D6751. Authors have tested only specific gravity, kinematic viscosity and heating value. Fuel mixtures can only be produced and analyzed after biodiesel quality is determined. As the obtained biodiesel has not been tested, it cannot be concluded that „The production of biodiesel from overripe fruits with the presence of pests is an alternative for the utilization of the waste generated”.
If tests are performed according to the requirements of the standards, it is not necessary to describe the method of analysis in the methodological part. There is no statistical data analysis.
Author Response
REVIEWER 1
- Idea of the research of the manuscript is interesting and relevant, however the manuscript is weak. The authors' contribution is not significant, as they did not optimize the biodiesel production process, but applied the optimal conditions chosen by other authors. However, it cannot be said that these conditions are the optimum because other authors used different raw materials (Zanthoxylum bungeanum and ilama seeds oils). If others authors’ obtained optimum conditions are used, esters must be tested for compliance with standards EN14214 and / or ASTM D6751. Authors have tested only specific gravity, kinematic viscosity and heating value. Fuel mixtures can only be produced and analyzed after biodiesel quality is determined. As the obtained biodiesel has not been tested, it cannot be concluded that „The production of biodiesel from overripe fruits with the presence of pests is an alternative for the utilization of the waste generated”.
Thanks for your comment.
The aim of this research was to determine, for the first time, the usefulness of P. schiedeana fruits (overripe, dehydrated and with the presence of pests) as a potential source to be transformed into biodiesel by alkaline transesterification and not the optimization of the conditions. As we mentioned earlier, this first study will serve as the baseline of the future work as the optimization of the extraction conditions and the determination of quality characteristics.
The next paragraph “The production of biodiesel from overripe fruits with the presence of pests is an alternative for the utilization of the waste generated” was replaced by “The production of biodiesel from overripe fruits with the presence of pests could become an alternative for the utilization of the residues generated. However, more studies are needed to confirm it”.
- If tests are performed according to the requirements of the standards, it is not necessary to describe the method of analysis in the methodological part. There is no statistical data analysis.
Thanks for your observation, however in the opinion of the authors, it is necessary to specify the methodology in detail to facilitate the reader's full understanding of the study.
Reviewer 2 Report
This manuscript presents very interesting results regarding the use of useful agricultural waste for obtaining oils, which can be used in the manufacture of biofuels. Given the current interest in obtaining biofuels from crops, not usable as food, and because the study is well carried out and adequately described, it can be accepted for publication in its current form.
Author Response
REVIEWER 2
This manuscript presents very interesting results regarding the use of useful agricultural waste for obtaining oils, which can be used in the manufacture of biofuels. Given the current interest in obtaining biofuels from crops, not usable as food, and because the study is well carried out and adequately described, it can be accepted for publication in its current form.
- Thank you for your positive comment. We worked persistently on this article to make it interesting and easy to read.

Reviewer 3 Report
The authors of the current manuscript describe the quality characteristics of biodiesel produced from waste from Persea schiedeana fruits. The manuscript is very interesting, but a very thorough review must be done before it is finally published in the journal.
- What problem did your research solve? What is your contribution to the advancement of this kind of research? Better explain the urgency of your research hypothesis.
- Literature review, in my point of view is weak, which required to improve and strengthen. The novelty of the work must be clearly addressed and discussed, compare your research with existing research findings and highlight novelty. Following paper can be cited: https://doi.org/10.3390/en10010130
- Build your research hypothesis more clearly (straightforward and groundbreaking claim that is confirmable or refutable) at the end of the Introduction chapter, justify the urgency of its investigation from industrial point of view.
- Section 2 should be Materials and Methods. The structure of the manuscript is not correct.
- Our readers should find a detailed description of all your procedures (step by step), anybody who reads this manuscript should be able to repeat your methods and obtain exactly the same results.
- Discuss the relevance of your work.
- It would be convenient for the manuscript to be reviewed by a native English speaker so that any grammatical errors it may contain are corrected.
Author Response
REVIEWER 3
The authors of the current manuscript describe the quality characteristics of biodiesel produced from waste from Persea schiedeana fruits. The manuscript is very interesting, but a very thorough review must be done before it is finally published in the journal.
- What problem did your research solve? What is your contribution to the advancement of this kind of research? Better explain the urgency of your research hypothesis.
Thanks for your suggestion. At the end of the introduction we have justified our research (lines 62-70): “Due to the significant growth of population and the increase of energy demand and the fact that fossil fuels are limited sources of energy, to date, some authors have tried to examine the potential of food waste [15,16] as avocado seed as an environmentally friendly alternative energy source [17–19]. However, the potential of P. schiedeana fruits for biodiesel production has not been evaluated so far, despite the considerable losses of these fruits attributed to their short postharvest life and pest attacks. Thus, the aim of this research was to determine, for the first time, the usefulness of P. schiedeana fruits (overripe, dehydrated and with the presence of pests) as a potential source to be transformed into biodiesel by alkaline transesterification”.
- Literature review, in my point of view is weak, which required to improve and strengthen. The novelty of the work must be clearly addressed and discussed, compare your research with existing research findings and highlight novelty. Following paper can be cited: https://doi.org/10.3390/en10010130
Thank you for your observation, however, in the opinion of all the authors, the comparison of the results obtained was made with studies using the same plant material and/or similar species such as avocado. The study you have proposed was used to enrich the introduction.
- Build your research hypothesis more clearly (straightforward and groundbreaking claim that is confirmable or refutable) at the end of the Introduction chapter, justify the urgency of its investigation from industrial point of view.
According to your suggestions in point 1, the hypothesis of this work was justified.
- Section 2 should be Materials and Methods. The structure of the manuscript is not correct.
Thank you for the observation, however, the structure of the manuscript is correct following the Instructions for Authors available in https://www.mdpi.com/journal/plants/instructions#preparation
- Our readers should find a detailed description of all your procedures (step by step), anybody who reads this manuscript should be able to repeat your methods and obtain exactly the same results.
The methodology has been clearly explained so as not to confuse readers. In fact another reviewer suggested that since the tests were performed according to the requirements of the standards, it was not necessary to describe the method of analysis in the methodological part. However, in our opinion we decided to leave the detailed information.
- Discuss the relevance of your work.
Thank you for your comment that we are sure contributes to improve the quality of this article. In the discussion section (lines 120-128) we have added more information.
- It would be convenient for the manuscript to be reviewed by a native English speaker so that any grammatical errors it may contain are corrected.
According to your suggestions, the article was sent for English revision by a native English speaker. Thank you again.
Round 2
Reviewer 1 Report
The manuscript is better after corrections, however it is not good enough, methods part seems artificially expanded to describe the methodologies that can be found in the standards. Authors says that it is necessary to specify the methods which were used to make it easier for readers to understand the research, however gravity, kinematic viscosity and heating value of the mixtures are not specified standards mentioned only. It is not clear under what temperature biodiesel synthesis was performed. Content of the moisture in the oil is very important factor to investigate possibility to use oil for biodiesel synthesis, why was it not determined? It is unclear why specific gravity, kinematic viscosity and heating value were determined and other parameter were not. Authors did not take note of the remark “There is no statistical data analysis”. There is no such a measurement “Kj·Kg-1”, kJ·Kg-1 should be.
Author Response
REVIEWER 1
- The manuscript is better after corrections, however it is not good enough, methods part seems artificially expanded to describe the methodologies that can be found in the standards. Authors says that it is necessary to specify the methods which were used to make it easier for readers to understand the research, however gravity, kinematic viscosity and heating value of the mixtures are not specified standards mentioned only.
Thanks for your observation. The following section has been removed from the article:
FFA: 0.5 g of oil was mixed with 10 mL of ether-ethanol (1:1) and 0.5 mL of phenolphthalein. The sample was stirred constantly and titrated with 0.1 N potassium hydroxide (KOH) until it turned a permanent pink color for at least 1 min. FFA were expressed as percent by mass according to the following equation:
|
|
(3) |
Where V is the volume of KOH used in the titration (mL), N is the exact normal of KOH, M is the molar mass per mole of oleic acid (g), m is the mass of the sample (g). The results were expressed as percentage of oleic acid per gram of sample (g).
Peroxide number: Oil (1 g) was mixed with 6 mL of glacial acetic acid/chloroform (3:2, v/v), the mixture was stirred and a saturated solution of potassium iodide (0.1 mL) was added, subsequently capped and allowed to stand for 1 min and finally distilled water (6 mL) was added. The titration was proceeded with a titrated solution of sodium thiosulfate (Na2S2O3) 0.01 N, during the titration the mixture was shaken vigorously and before the disappearance of the yellow color 0.1 mL of starch indicator solution was added, the titration was continued until the disappearance of the blue color in the mixture. A reagent blank was titrated in the same way as the sample. The reading was taken according to the milliliters spent in the titration and the peroxide value was obtained from the following equation:
|
|
(4) |
Where A is the milliliters of Na2S2O3 solution spent in the titration of the sample, A1 is the milliliters of Na2S2O3 solution spent in the titration of the blank, N is the normal of the Na2S2O3 solution, m is the weight of the sample (g). The results were expressed as milliequivalents of active oxygen per kilogram of oil (meq O·kg-1).
Iodine value: 0.2 g of oil were mixed with 10 mL of methylene chloride (CH2Cl2) and 10 mL of Wijs' reagent. The mixture was stirred and placed in the dark for 30 min, then 10 mL of potassium iodide (10 %) was added and titrated with Na2S2O3 0.1 N titrated solution, during the titration the mixture was stirred constantly. The sample was titrated until it turned straw yellow and then 1 mL of starch indicator solution was added, titrating until it turned from blue to colorless. At the same time a reagent blank was titrated in the same way as the sample. The reading was taken according to the milliliters spent in the titration and the iodine value was obtained from the following equation:
|
|
(5) |
Where V1 is the milliliters spent in the titration of the blank, V2 is the milliliters spent in the titration of the sample, N is the normal of the Na2S2O3 solution, m is the mass of the sample (g). The results were expressed as grams of iodine absorbed per 100 g of sample (g· 100 g-1).
Saponification number: 5 g of oil were mixed with 50 mL of 0.5 N KOH in a ball flask. The mixture was refluxed for 30 min with constant stirring, at the end of the hydrolysis reaction, the mixture was allowed to cool and 1 mL of phenolphthalein was added. The excess KOH was titrated with 0.5 N hydrochloric acid (HCl). A reagent blank was carried out with the same amounts used in the samples. The reading was taken according to the milliliters spent in the titration and the saponification index was obtained from the following equation
|
|
(6) |
Where V is the milliliters of HCl spent in the titration of the sample, B is the milliliters of HCl spent in the titration of the blank, N is the normal of HCl, m is the mass of the sample (g), 56.1 molecular weight of KOH (g mol-1). Results were expressed as milligrams of KOH required to saponify one gram of sample (mg·g-1).
- It is not clear under what temperature biodiesel synthesis was performed.
The biodiesel preparation was carried out at 60 °C. This information has been added (line 293).
- Content of the moisture in the oil is very important factor to investigate possibility to use oil for biodiesel synthesis, why was it not determined?.
Although the percentage of moisture was not determined in the oil obtained, as explained in the materials and methods section, the oil obtained was dried with anhydrous sodium sulfate to ensure that it was free of moisture. Furthermore, the absence of moisture was confirmed with the NMR spectrum since the HOD and H2O signals that would be indicative of moisture were not observed.
- It is unclear why specific gravity, kinematic viscosity and heating value were determined and other parameter were not.
Other parameters such as: cloud point, cetane index, flash point and pour point were not determined because there are no instruments to measure these properties. Furthermore, this study represents a starting point for future research.
- Authors did not take note of the remark “There is no statistical data analysis”.
The chemical properties of the oil were analyzed in triplicate, according to the methods of the International Organization for Standardization (ISO). In table 1 the standard deviation values (red color) have been added.
Regarding to fatty acids, the samples were not analyzed in triplicate because were determined based on the certificate method ISO 9001:2008 RSGC 238 .
Respect to values for mechanical properties and calorific value of oil, biodiesel and its blends. The results obtained are only for a determination. However, in other published studies, only one data has also been reported for these determinations:
- Hiwot, T. (2017). Determination of oil and biodiesel content, physicochemical properties of the oil extracted from avocado seed (Persea Americana) grown in Wonago and Dilla (gedeo zone), southern Ethiopia. Chem. Int, 3(3), 311-319.
- Dagde, K. K. (2019). Extraction of vegetable oil from avocado seeds for production of biodiesel. Journal of Applied Sciences and Environmental Management, 23(2), 215-221.
- Rajesh Kana, S., & Shaija, A. (2020). Performance, combustion and emission characteristics of a diesel engine using waste avocado biodiesel with manganese-doped alumina nanoparticles. International Journal of Ambient Energy, 1-8.
- Sathish, S., Narendrakumar, G., Vaithyasubramanian, S., & Sinduri, E. (2021). Mechanistic model for the batch extraction of oil from avocado seeds available for biofuel production. International Journal of Green Energy, 1-13.
- Anawe, P. A. L., & Adewale, F. J. (2018). Data on physico-chemical, performance, combustion and emission characteristics of Persea Americana Biodiesel and its blends on direct-injection, compression-ignition engines. Data in brief, 21, 1533-1540.
- There is no such a measurement “Kj·Kg-1”, kJ·kg-1 should be. Thanks for your comment, units were corrected.

Reviewer 3 Report
The authors have revised the manuscript in accordance with the reviewers' comments. The paper may be recommended for publication as it meets most of requirements to such type of articles.
Author Response
REVIEWER 3
- The authors have revised the manuscript in accordance with the reviewers' comments. The paper may be recommended for publication as it meets most of requirements to such type of articles.
Thank you for your positive comment. We worked persistently on this article to make it interesting and easy to read.
